# Skin Toxicity as a Predictor of Survival in Metastatic Colorectal Cancer Patients Treated with Anti-EGFR: Fact or Fallacy?

**DOI:** 10.3390/cancers15061663

**Published:** 2023-03-08

**Authors:** Ting-Yu Chiang, Hung-Chih Hsu, Yih-Jong Chern, Chun-Kai Liao, Yu-Jen Hsu, Wen-Sy Tsai, Pao-Shiu Hsieh, Yu-Fen Lin, Hsiu-Lan Lee, Jeng-Fu You

**Affiliations:** 1Department of Nursing, Chang Gung Memorial Hospital at Linkou, College of Medicine, Chang Gung University, Taoyuan 33305, Taiwan; 2Division of Hematology-Oncology, Chang Gung Memorial Hospital at Linkou, College of Medicine, Chang Gung University, Taoyuan 33305, Taiwan; 3Division of Colon and Rectal Surgery, Department of Surgery, Chang Gung Memorial Hospital at Linkou, College of Medicine, Chang Gung University, Taoyuan 33305, Taiwan

**Keywords:** metastatic CRC, mCRC, skin toxicity, survival, anti-EGFR

## Abstract

**Simple Summary:**

Targeted therapy and chemotherapy are the mainstays of treatment to improve the survival of patients with metastatic colorectal cancer (mCRC). When RAS and BRAF genes are normal, a molecular target treatment with an anti-EGFR antagonist is prescribed. Anti-EGFR antagonists induce skin responses in 50–70% of patients. There is an ongoing debate about whether the severe skin reactions brought on by anti-EGFR antagonists are associated with overall survival (OS) and progression-free survival (PFS). mCRC patients who received anti-EGFR therapy between October 2017 and October 2018 were retrospectively evaluated. Treatment with an anti-EGFR medication in the first-line setting was significantly associated with OS and PFS. In grades 1 and 2, there was no difference in the incidence of acne between males and females, although in grades 3 and 4, males were at a higher risk than females. In this study, skin toxicity did not predict the effectiveness of the anti-EGFR medication.

**Abstract:**

The primary treatment for metastatic colorectal cancer (mCRC) consists of targeted therapy and chemotherapy to improve survival. A molecular target drug with an anti-epidermal growth factor receptor (EGFR) antagonist is recommended when the RAS and BRAF genes are normal. About 50–70% of patients using anti-EGFR antagonists will experience skin reactions. Some studies have shown that severe skin reactions caused by anti-EGFR antagonists may be linked to overall survival (OS) and progression-free survival (PFS), but the results are still uncertain. These data of mCRC patients who underwent anti-EGFR therapy between October 2017 and October 2018 were analyzed retrospectively. A total of 111 patients were included in this study. The survival results showed that gender, age, body mass index, primary tumor site, and recurrence did not significantly affect OS and PFS. However, the first-line anti-EGFR inhibitor treatment was significantly associated with OS (*p* < 0.001) and PFS (*p* < 0.001). There was no significant difference in the incidence of acne between males and females in grades 1 and 2, while males have a greater risk in grades 3 and 4 than females (20.3 vs. 4.8%; *p*-value = 0.041). Skin toxicity was not a predictor of anti-EGFR treatment response in this investigation.

## 1. Introduction

In 2020, 1.93 million new diagnoses of colorectal cancer (CRC) were given, making it the fourth most common cancer in the world, and the third leading cause of cancer-related deaths [1]. In Taiwan, CRC is the second most common cancer and has the third-highest mortality rate [2]. Stage 4 colorectal cancer accounts for 20–25% of all cases, andthe survival rate is 15.1% [3]. Metastatic colorectal cancer most commonly metastasizes to the liver, lung, and peritoneum. The primary treatment of metastatic CRC is targeted therapy combined with chemotherapy to improve survival. Additionally, recent research has shown that applying the Oct4–Sox2 transcription factor decoy strategy to cancer stem-like cells, or combinations of curcumin and chemotherapy, may have an anti-cancer effect [4,5,6]. The choice of a molecular target drug must depend on the status of the patient’s RAS (including KRAS and NRAS) and BRAF genes. Anti-epidermal growth factor receptor (anti-EGFR) drugs can only be recommended when both RAS and BRAF genes are wild type, and cetuximab or panitumumab are suggested in this case according to NCCN guidelines [7]. However, if RAS or BRAF are genetically mutated, an anti-vascular endothelial growth factor (anti-VEGF) drug such as bevacizumab is an option.

Approximately 50–70% of patients with metastatic CRC have a wild-type RAS gene, and anti-EGFR antagonists are recommended based on evidence in treating CRC worldwide [8]. However, utilizing anti-EGFR antagonists is inevitably accompanied by some additional side effects. EGFRs expressed in normal human skin tissue are responsible for cell growth, the proliferation of keratinocytes, and the production of the outer root sheath of hair follicles. Skin reactions are common adverse reactions to anti-EGFR therapy, reported in up to 45–100% of patients, including acneiform rash, dry skin, and paronychia [9,10].

The skin reactions not only cause psychological and symptomological distress because of the changes in appearance, but also affect patients’ quality of life [11]. The management of skin reactions has been proposed, namely via pre-emptive treatment such as the STEPP (Skin Toxicity Evaluation Protocol with Panitumumab) and J-STEPP (randomized controlled trial on the skin toxicity of panitumumab in Japanese patients with metastatic colorectal cancer), which showed that pre-emptive treatment can reduce skin reactions caused by EGFR treatment [12,13,14]. Some retrospective data and randomized trials have shown that the severe skin reactions caused by anti-EGFR antagonists correlate with overall survival (OS) and progression-free survival (PFS), but the result is still controversial [15,16,17]. In addition, another issue regarding gender differences in the efficacy and toxicity of anti-EGFR therapy is the limited available data [18].

Information on the relationship of OS and PFS with anti-EGFR antagonist-induced skin reactions is limited and contentious. In the present study, we aimed to investigate the association of OS and PFS with anti-EGFR antagonist-induced skin reactions according to age and gender.

## 2. Material and Methods

### 2.1. Design and Participants

We conducted a retrospective analysis by reviewing the clinicopathological data and the follow-up outcomes of mCRC patients who received anti-EGFR therapy at the Colorectal and Oncology Division at the Chang Gung Memorial Hospital of Linkou Branch between October 2017 and October 2018 [11]. The Institutional Review Board approved this study (IRB No.202202367B0) through the Chang Gung Memorial Hospital of Linkou Branch. All patients who signed informed consent were included in this study.

Patients who met the following criteria were enrolled: (a) histologically confirmed colorectal adenocarcinoma; (b) metastases (stage IV CRC) or recurrent CRC after curative resection determined by imaging or pathologically; (c) underwent chemotherapy combined with anti-EGFR therapy and received at least two to four treatment doses; and (d) physical performance was less than 60 on the Karnofsky Performance Status Scale (KPS) [19]. Patients were excluded if treatment was discontinued due to disease progression or intolerance to therapy or if they died early during treatment.

### 2.2. Clinicopathological Variables and Measurement Outcomes

Baseline characteristics included age, gender, body mass index (BMI), KPS, primary tumor location, line of therapy, chemotherapy after surgery, and recurrence. Results of EGFR expression, including composite score (cite the reference) and skin reactions, were also collected. All patients receiving anti-EGFR therapy had regularly scheduled outpatient clinic visits or admissions and were assessed for treatment response by CT scans of the chest or abdomen every 3 months, or serum tumor markers. The treatment response assessment used the Response Evaluation Criteria in Solid Tumors [20,21,22].

The outcome measures for efficacy endpoints were PFS and OS in this study. PFS was defined as the interval from the first anti-EGFR treatment to the first progression, death, or final follow-up date. OS was defined as the interval from the first anti-EGFR treatment to the date of death or last follow-up.

### 2.3. Statistical Analyses

All analyses were conducted using SPSS Statistics v.26 (IBM Corp., Armonk, NY, USA). An α level of 0.05 was considered statistically significant, and all *p*-values were two-sided. The characteristics of baseline EGFR expression, results of the composite scores, and Anti-EGFR Skin Reactions were analyzed with categorical variables, presented as frequencies and proportions, and were compared using the X^2^ test. Continuous variables are expressed as means and standard deviations and were analyzed using the Student *t*-test and One-Way ANOVA test. Survival was analyzed using the Kaplan–Meier method. Cox proportional hazards regression analysis was used to calculate hazard ratios (HRs) with 95% confidence intervals.

## 3. Result

### 3.1. Patients’ Characteristics

Between October 2017 and October 2018, 111 patients with mCRC underwent anti-EGFR treatment and were included in this study. The demographics and clinical characteristics are listed in Table 1. There were 69 male (62.2%) and 42 female (37.8%) patients. The mean age was 58.9 (SD ± 13.29; range 31–90) years. A significantly higher number of male patients were diagnosed with colon cancer (n = 69; 62.2%), and 73 patients (male and female combined) were treated with first-line anti-EGFR medication (65.8%). The other demographic and clinical characteristics are listed in Table 1.

### 3.2. EGFR Expression Levels in CRCs and Skin Reactions with Anti-EGFR Inhibitor

In this study, EGFR expression was identified in 102 of 111 individuals. The results of the composite score, which is the product of the positive grade multiplied by the labeling-intensity score, are presented in Table 2. EGFR expression [23] was classified as high (>=6) in 60 (58.9%) patients and low (<6) in 42 (41.2%) patients.

Table 3 shows that most acne patients who had a skin reaction to the anti-EGFR inhibitor were grade 0–2 (n = 95, or 85.6%) and most paronychia patients were grade 0–1 (n = 91, or 82%).

### 3.3. Association of Variables with OS and PFS

Table 4 and Table 5 display the survival results of the study, which revealed that gender, age, BMI, primary tumor site, and recurrence did not significantly affect OS and PFS. However, the first-line anti-EGFR inhibitor treatment was significantly associated with OS (*p* < 0.001) and PFS (*p* < 0.001).

The average OS for patients who received anti-EGFR inhibitor therapy as the first-line was 38.9 (SD ± 2.644) months, and the average OS for patients who received anti-EGFR inhibitor therapy as the third-line was 19.8 (SD ± 2.333) months, with a *p*-value of 0.001 (Figure 1a and Figure 2a). First-line anti-EGFR inhibitor patients had a mean PFS of 20.9 months (SD ± 1.967), while third-line patients had a mean PFS of 10.1 months (SD ± 1.375) (Figure 1b and Figure 2b).

There was no statistically significant difference between OS and PFS and the assessment of the EGFR composite score, skin responses, and paronychia (Figure 3).

### 3.4. Association of Skin Reactions with Age and Gender

#### 3.4.1. Acne with Age and Gender

These findings revealed a statistically significant difference between age and the severity of acne caused by anti-EGFR inhibitors (*p* < 0.001) (Table 6).

The mean age of the 23 patients with grade 0 acne was 66.7 (SD ± 14.138); the number of patients with acne grades 1–2 was 72, with a mean age of 59.1 (SD ± 11.712); the number of patients with acne grades 3–4 was 16, with a mean age of 46.9 (SD ± 10.329). The results revealed that skin acne was more severe in individuals with lower mean ages and vice versa. Figure 4a also shows the tendency for skin acne to become less severe as people get older.

Male patients who received anti-EFGR had significantly more skin acne than female patients (*p*-value = 0.041) (Table 6). In particular, the incidence of skin acne in grades 3–4 is higher in men than in women (14/69; 20.3% vs. 2/42; 4.8%) (Figure 4b).

#### 3.4.2. Paronychia with Age and Gender

Age had no significant impact on the severity of anti-EGFR inhibitor-induced paronychia (*p* = 0.818). Likewise, there was no statistically significant difference in the incidence rate of paronychia across genders (*p*-value = 0.746) (Table 6).

The receiver operating characteristic (ROC) curve elucidating the prediction of age and skin acne greater than grade 2 is shown for the entire cohort of patients. As displayed in Figure 5, the area under the curve (AUC) was 0.705 (95% CI: 0.601–0.809; *p* < 0.001).

## 4. Discussion

This study demonstrated that skin toxicity is not a predictor of OS and PFS in patients with metastatic colorectal cancer treated with anti-EGFR inhibitors. Most patients treated with anti-EGFR inhibitors experienced only mild paronychia and skin reactions ranging from grades 0 to 2. Furthermore, regardless of gender, age, BMI, primary tumor site, recurrence, composite score, skin response, or paronychia, only the first-line treatment being an anti-EGFR inhibitor administered for mCRC had a significant influence on OS and PFS. The research examined the effects of gender, age, and acne severity in patients treated with anti-EGFR inhibitors and found that male patients and younger age groups were more likely to experience a higher incidence of severe acne due to the treatment. This result emphasizes the importance of considering individual patient characteristics when administering anti-EGFR inhibitors to reduce the risk of adverse skin reactions. Moreover, the study shows that male patients have a higher incidence of grades 3–4 acne than females. The evidence indicates that a higher incidence of skin rashes in males might be attributed to the effects of a lack of skincare, male hormones, and the secretion of anti-EGFR antibodies in sweat [24].

Nonetheless, some other research has reached the opposite conclusion. It is not uncommon in a clinical setting for physicians to presume that a patient’s anti-EGFR medication-induced sensitive skin inflammatory response is a predictor of survival or effectiveness of disease treatment, or even that more severe skin reactions are more likely to survive than minor symptoms. A post hoc analysis from the CAVE mCRC study demonstrated that the occurrence of grade 2–3 skin toxicity is a strong predictor of patient outcome [15]. The FIRE-3 trial showed that both the cetuximab-induced skin toxicity and early tumor shrinking were independent predictors of OS [25]. Tougeron D. et al., obtained skin biopsies from mCRC patients treated with cetuximab to examine what produced the rash and how it affected treatment. They found that Th2-related and keratinocyte-derived cytokines in the skin were correlated with anti-EGFR efficacy [16]. Cetuximab-induced skin toxicity of grades 2–4 has also been linked to a shorter OS in patients with recurrent or metastatic head and neck squamous cell carcinoma [26]. A recent study showed that a lower incidence of any grade skin reactions in the pre-emptive compared to the reactive cohort did not reduce the efficacy of antineoplastic therapy compared to the reactive in terms of ORR, median PFS, and median OS [27]. In this study, however, there was no statistically significant association between the degree of skin toxicity and OS or PFS in patients with mCRC who were administered cetuximab.

This study showed that patients younger than 70 and male had a higher incidence of severe cetuximab-induced skin reactions, which is comparable with the findings of previous studies [17,28]. Patients over the age of 70 showed relatively less skin toxicity, possibly due to the prolonged growth and cell metabolism cycles of the skin with age, slowed epidermal proliferation, reduced collagen synthesis, and fewer microorganisms on the surface of the skin than in younger patients, which leads to reasonably mild skin inflammation [29,30]. According to clinical evidence, men are less likely than women to care for their skin. Therefore, even when the clinical medical staff educate patients about skin care, male patients often have a poorer compliance rate and are less aware of how to choose washing and skin care products. Inadequate skin care practices, such as improper washing and moisturizing, might interfere with normal metabolic function and worsen skin inflammation.

Although anti-EGFR inhibitors have a remarkable survival rate for mCRC patients with wild-type RAS, the associated skin toxicity and physical and mental image change negatively impact the quality of life [12,31,32,33]. Practical remedies and appropriate skin symptom care can increase patients’ adherence to treatment and overall life satisfaction [12,34,35,36]. Antibiotics used prophylactically, gentle washing methods, increased skin moisture, and chemical and physical sun protection can effectively reduce skin inflammation and irritation while receiving treatment [9,37,38]. In addition, some diet-related studies have found that apigenin can inhibit inflammation and act as an antioxidant [39], and vitamin E can shield cell membranes from lipid peroxidation, avoiding oxidative skin damage [40]; dark green vegetables and nuts provide significant quantities of vitamin E and beneficial micro-organisms. Prebiotic fermentation in the gut microbiota can also result in the synthesis of short-chain fatty acids, which may increase epithelial barrier function and regulate the inflammatory response [41].

Based on the results mentioned above, we offer a few potential explanations for why the link between skin toxicity and PFS and OS in patients with mCRC receiving anti-EGFR inhibitors is still unclear. Firstly, the inconsistency could result from ethnic variances in skin characteristics and geography-related variations. Furthermore, the intensity of an anti-EFGR inhibitor-induced skin reaction may vary depending on the research study due to a variety of known and unknown confounders, including age, sex, and other factors. Additionally, the severity of skin toxicity might be lessened if pre-emptive skin care interventions differ based on the studies and the patient administering anti-EGFR therapies [13,14].

## 5. Limitations

This study has several limitations. First, this retrospective study may introduce unavoidable biases. Second, the confounding factors influencing skin toxicity may not be properly adjusted, even with multivariable analysis. Third, the hot and humid climate on Taiwan’s subtropical island may affect skin toxicity during treatment. Fourth, personal skin-care habits should have an impact on the development of skin toxicity. Fifth, this study utilized a pre-emptive approach in initiating anti-EFGR inhibitors to instruct patients on how to take care of their skin, which could have altered the outcomes. Pre-emptive interventions include giving oral antibiotics, promoting proper skin care, emphasizing washing and moisturizing, sun protection, a balanced diet and stress management, enhancing self-confidence in one’s appearance, and supporting adherence to treatments.

Despite the limitations of this study, it also presents crucial insights and valuable information that can be used to inform future research and clinical practice. One common misconception among clinicians is that severe skin acne in patients receiving anti-EGFR inhibitors indicates better treatment outcomes and improved survival rates, which have been widely circulated and followed by some practitioners. The current study effectively refutes this misconception and provides clear evidence that severe acne caused by anti-EGFR inhibitors is, in fact, a sign of adverse reactions to the medication. The study’s findings highlight the importance of monitoring patients for skin reactions during anti-EGFR inhibitor treatment and the need for individualized treatment plans that account for patient characteristics, such as age and gender. By dispelling this fallacy, the research contributes to the expanding body of the literature on the use of anti-EGFR inhibitors and highlights the significance of evidence-based medicine in clinical practice.

## 6. Conclusions

In conclusion, despite the limitations of a single-institute retrospective study, our findings did not show skin toxicity as a predictor of response to anti-EGFR treatment. Most patients with skin reactions after the pre-emptive strategy who received an anti-EGFR inhibitor had milder grades of colorectal cancer in this study. Only the use of an anti-EGFR agent as a first-line treatment for metastatic colorectal cancer significantly influenced survival results. More robust methods with larger sample sizes may yield a more reliable outcome.

## Figures and Tables

**Figure 1 cancers-15-01663-f001:**
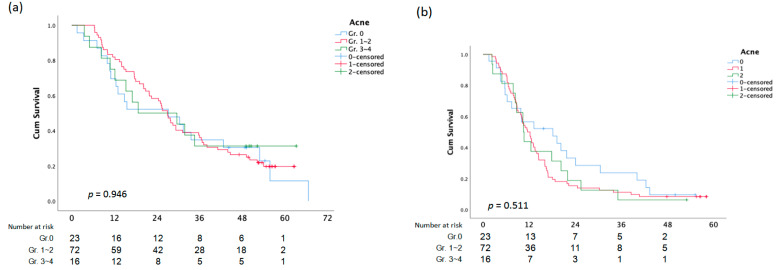
Overall Survival (**a**)and PFS (**b**) analysis according to Acne.

**Figure 2 cancers-15-01663-f002:**
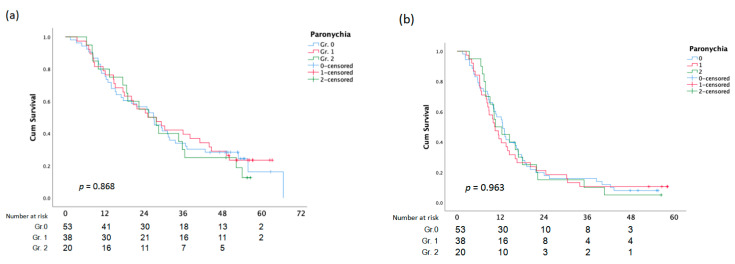
Overall Survival (**a**)and PFS (**b**) analysis according to Paronychia.

**Figure 3 cancers-15-01663-f003:**
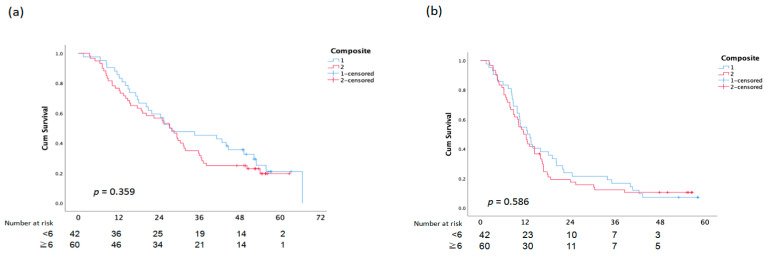
Overall Survival (**a**) and PFS (**b**) analysis according to Composite Score.

**Figure 4 cancers-15-01663-f004:**
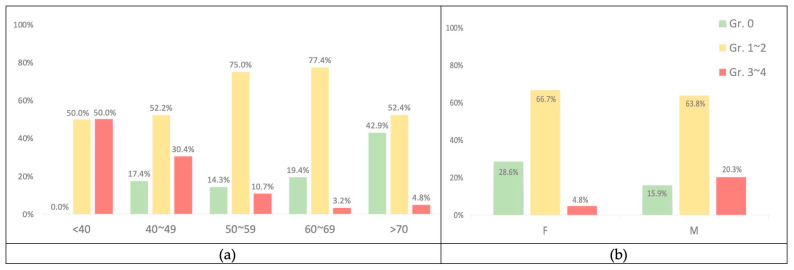
(**a**) The percentage according to age with acne. (**b**) The percentage according to gender with acne.

**Figure 5 cancers-15-01663-f005:**
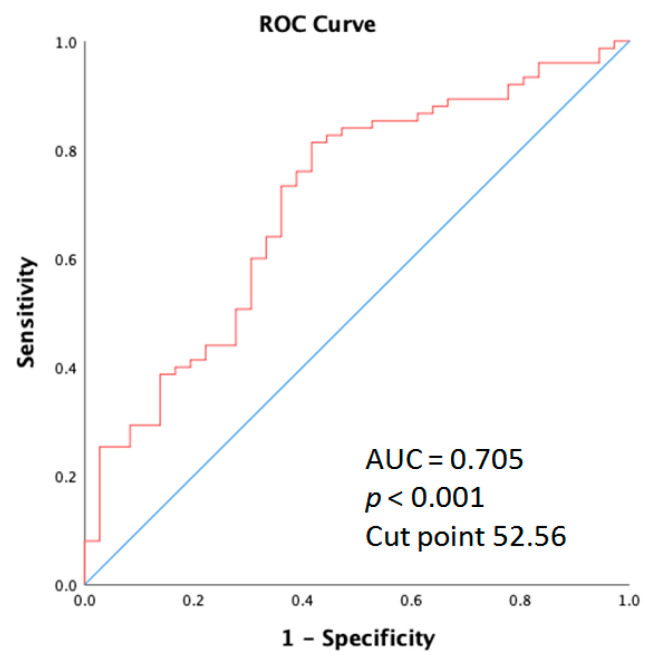
Receiver operation characteristic (ROC) curve analysis of the age for acne in metastatic colorectal cancer patients who received anti-EGFR inhibitor therapy.

**Table 1 cancers-15-01663-t001:** Characteristics of patients at baseline (N = 111).

Variables	n	%
Gender		
Female	42	37.8%
Male	69	62.2%
Age (mean ± SD, range) years 58.88 ± 13.29, 30.6–90.29		
<40	8	7.2%
40~49	23	20.7%
50~59	28	25.2%
60~69	31	27.9%
>70	21	18.9%
KPS		
60	2	1.8%
70	10	9%
80	6	5.4%
90	93	83.8%
BMI (kg/m^2^) mean ± SD		24.23 ± 4.24
≤25	71	64%
>25	40	36%
Primary site		
Colon	69	62.2%
Rectum	42	37.8%
Line Therapy		
1	73	65.80%
3	38	34.20%
Chemotherapy after Surgery		
Yes	20	18%
No	91	82%
Recurrence		
No	47	42.3%
Yes	64	57.7%

KPS: Karnofsky Performance Scale; BMI: body mass index.

**Table 2 cancers-15-01663-t002:** EGFR expression: results of the composite scores (N = 111).

Score	*n*	%
1	19	18.6%
2	12	11.8%
3	6	5.9%
4	5	4.9%
6	45	44.1%
9	15	14.7%
Total	102	100.0%
Missing	9	

**Table 3 cancers-15-01663-t003:** Anti-EGFR Skin Reactions (N = 111).

Variable	n	%
Acne		
Grade 0	23	20.7%
Grade 1	52	46.8%
Grade 2	20	18.0%
Grade 3	15	13.5%
Grade 4	1	0.9%
Paronychia		
Grade 0	53	47.7%
Grade 1	38	34.2%
Grade 2	20	18.0%

**Table 4 cancers-15-01663-t004:** Association of clinical variables and skin reactions with median overall survival by univariate analysis (N = 111).

	Overall Survival	Univariate		
Variable	Mean ± SD	HR	95% CI	*p*-Value
Gender				
Female	33.495 ± 3.449	1		
Male	30.258 ± 2.444	1.175	0.757–1.825	0.473
Age				
<40	42.951 ± 7.976	1		
40~49	31.318 ± 4.556	1.886	0.634–5.611	0.254
50~59	36.626 ± 4.295	1.597	0.546–4.677	0.393
60~69	25.633 ± 2.946	2.574	0.894–7.404	0.08
>70	26.333 ± 3.43	2.728	0.923–8.062	0.069
BMI				
≤25	34.177 ± 2.707	1		
>25	28.339 ± 3.04	1.342	0.87–2.069	0.183
Primary Site			-	
Colon	30.573 ± 2.613	1	-	
Rectum	34.301 ± 3.259	0.836	0.534–1.307	0.431
Line Therapy			-	
1	38.889 ± 2.644	1	-	
3	19.739 ± 2.333	2.881	1.86–4.464	<0.001
Recurrence			-	
No	34.662 ± 3.036	1	-	
Yes	29.931 ± 2.705	1.343	0.868–2.076	0.185
CS			-	
<6	35.819 ± 3.481	1	-	
≥6	30.495 ± 2.637	1.237	0.784–1.952	0.360
Skin reaction			-	
Acne			-	
Grade 0	30.316 ± 4.833	1	-	
Grade 1–2	32.105 ± 2.324	0.931	0.547–1.582	0.791
Grade 3–4	31.904 ± 5.85	0.888	0.419–1.881	0.756
Paronychia			-	
Grade 0	31.633 ± 3.093	1	-	
Grade 1	32.723 ± 3.451	0.941	0.583–1.519	0.803
Grade 2	29.785 ± 3.882	1.107	0.628–1.954	0.725

CS: Composite Score; BMI: body mass index; HR: hazard ratio.

**Table 5 cancers-15-01663-t005:** Association of clinical variables and skin reactions with median Progression Free Survival by univariate analysis (N = 111).

		Univariate		
Variable	Mean ± SD	HR	95% CI	*p*-Value
Gender				
Female	18.428 ± 2.481	1		
Male	16.292 ± 1.746	1.14	0.762–1.707	0.523
Age				
<40	11.512 ± 2.469	1		
40~49	15.825 ± 2.474	0.772	0.343–1.733	0.53
50~59	19.25 ± 3.309	0.6	0.267–1.345	0.215
60~69	19.443 ± 3.275	0.602	0.269–1.343	0.215
>70	14.05 ± 2.154	0.833	0.366–1.895	0.663
BMI				
≤25	18.765 ± 1.951	1		
>25	14.393 ± 2.023	1.363	0.909–2.043	0.134
Primary Site				
Colon	17.988 ± 1.977	1		
Rectum	16.557 ± 2.246	1.107	0.738–1.662	0.623
Line Therapy				
1	20.889 ± 1.967	1		
3	10.064 ± 1.375	2.666	1.751–4.06	<0.001
Recurrence				
No	18.838 ± 2.293	1		
Yes	16 ± 1.875	1.287	0.865–1.916	0.213
CS				
<6	18.813 ± 2.404	1		
≥6	16.773 ± 2.007	1.122	0.741–1.7	0.586
Skin reaction				
Acne				
Grade 0	20.706 ± 3.563	1		
Grade 1–2	16.231 ± 1.728	1.342	0.81–2.223	0.253
Grade 3–4	15.788 ± 3.218	1.311	0.67–2.567	0.429
Paronychia				
Grade 0	17.27 ± 2.066	1		
Grade 1	16.931 ± 2.599	1.062	0.684–1.65	0.788
Grade 2	16.74 ± 2.902	1.038	0.61–1.767	0.891

**Table 6 cancers-15-01663-t006:** Association between skin reactions with Age and Gender (N = 111).

Variable	Age				Gender		
	n	Mean ± SD		*p*-Value	n (%)		*p*-Value
			95% CI		F	M	
Acne				<0.001			0.041
Grade 0	23	66.681 ± 14.138	60.567–72.795		12 (28.6%)	11 (15.9%)	
Grade 1–2	72	59.048 ± 11.712	56.295–61.800		28 (66.7%)	44 (63.8%)	
Grade 3–4	16	46.932 ± 10.329	41.427–52.436		2 (4.8%)	14 (20.3%)	
Paronychia				0.818			0.746
Grade 0	53	59.574 ± 13.972	55.723–63.425		22 (52.4%)	31 (44.9%)	
Grade 1	38	58.714 ± 11.93	54.793–62.635		13 (31.0%)	25 (36.2%)	
Grade 2	20	57.372 ± 14.391	50.637–64.108		7 (16.6%)	13 (18.8%)	

## Data Availability

Due to privacy and ethical concerns, data details and how to request access are available from the corresponding author.

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
