# Peer review of "Skin Toxicity as a Predictor of Survival in Metastatic Colorectal Cancer Patients Treated with Anti-EGFR: Fact or Fallacy?"

_cancers, 2023, doi:10.3390/cancers15061663_

Round 1

Reviewer 1 Report

Comments  

1-      The entire manuscript requires some English editing.

2-      Please improve the quality of Figure 4.

3-      In the introduction section, try to add information regarding novel targeted cancer therapy methods in colorectal cancer stem cell elimination. For this purpose, the authors are advised to use the following studies:

DOI: 10.2217/nnm-2021-0334

doi.org/10.1007/s11033-020-05280-2

DOI: 10.1007/s11033-020-05737-4

Author Response

As attachment.

Reviewer 2 Report

The current study retrospectively determined the relationship of OS and PFS with first line anti-EGFR antagonist-induced skin reactions according to age and gender. The study fits in the preview of the journal and can be published after addressing the concerns.

Almost 30% of the cited references are older than five years.

Under the patient characteristics, the following modified statement or similar needs to be replaced with the current sentence to provide a correct view: A significantly higher number of 'male' patients were diagnosed with colon cancer (n = 69; 62.2%) and 73 patients (male and female combined) were treated with first-line anti-EGFR medication (65.8%).

The authors, under table 5, mentioned that skin reactions were worse in the younger patients and less severe in the older ones. The authors are requested to provide statistical significance of the comparison. Only eight of the 111 patients in the study are less than 40 years old. The authors are asked to redefine the patient group (young vs. old) or provide the younger vs. older age group definition, as the current impression provided (young vs. old) can be misleading,

Similarly, under figure 4, describing incidence of acne, the authors mention: "males showed a greater risk in grades 3 and 4 than females (20.3 vs. 4.8%; p-value = 0.041)". The number of males involved in grade 3 and 4 are 14, while only two females are present. How did the authors obtain statistical significance with just two females? Also, the authors need to clarify this vast difference in the study population immediately after the quoted sentence.

Also, under discussion, the authors states: "An investigation of gender, age, and the severity of acne treated with anti-EGFR inhibitors indicated that younger age groups had more severe acne, whereas older age groups had less severe acne." The provided statement can be misleading under the context of the queries listed above. The authors can rather state the ages and the presence or absence of significance.

Likewise, the following sentence in discussion may have to be re-addressed: "Moreover, men are more likely than women to develop acne, and men have a higher incidence of grades 3–4 acne than women."

Author Response

As attachment.

Reviewer 3 Report

In the manuscript titled: “Skin Toxicity as a Predictor of Survival in Metastatic Colorectal Cancer Patients Treated with Anti-EGFR: Fact or Fallacy?” the authors present a retrospective study evaluating the occurrence of acne and paronychia. The authors report no statistically significant association between signs of skin toxicity and survival (overall survival and progression-free survival). The authors report a greater incidence of acne in male patients with grade 3 and 4 colorectal cancer (p-value 0.041): and a tendency for worsening skin reactions in younger patients with no statistical analysis. The design of the tables is not clear, and the information is properly organized (E.g., in Table 3, in the Variable column, Acne and Paronychia are not aligned). The text size in Figure 2 and Figure 4 is too small to be properly read, even in an electronic format. Although valuable, the information from the cohort collected from the authors does not provide enough new information or helpful observations to be published as an original research paper.

Author Response

As attachment.

Round 2

Reviewer 3 Report

The authors improved the quality of the figures and presentation of the results in the manuscript. The relevance of the study is now clear from the discussion of their results.